# The Effect of Manufacture Process on Mechanical Properties and Burning Behavior of Epoxy-Based Hybrid Composites

**DOI:** 10.3390/ma15010301

**Published:** 2022-01-01

**Authors:** Kamila Sałasińska, Peteris Cabulis, Mikelis Kirpluks, Andrejs Kovalovs, Paweł Kozikowski, Mateusz Barczewski, Maciej Celiński, Kamila Mizera, Marta Gałecka, Eduard Skukis, Kaspars Kalnins, Ugis Cabulis, Anna Boczkowska

**Affiliations:** 1Faculty of Materials Science and Engineering, Warsaw University of Technology, Wołoska 141, 02-507 Warsaw, Poland; sgalecki44@gmail.com (M.G.); anna.boczkowska@pw.edu.pl (A.B.); 2Department of Chemical, Biological and Aerosol Hazards, Central Institute for Labour Protection—National Research Institute, Czerniakowsa 16, 00-701 Warsaw, Poland; pawel.kozikowski@ciop.pl (P.K.); maciej.celinski@ciop.pl (M.C.); kamila.mizera@ciop.pl (K.M.); 3Institute of Materials and Structures, Riga Technical University, 6b Kipsalas St., 1048 Riga, Latvia; peteris@ritols.lv (P.C.); andrejs.kovalovs@rtu.lv (A.K.); edskukis@gmail.com (E.S.); kaspars.kalnins@rtu.lv (K.K.); 4Polymer Laboratory, Latvian State Institute of Wood Chemistry, 27 Dzerbenes St., 1006 Riga, Latvia; mkirpluks@kki.lv (M.K.); cabulis@kki.lv (U.C.); 5Institute of Materials Technology, Poznan University of Technology, Piotrowo 3, 61-138 Poznan, Poland; mateusz.barczewski@put.poznan.pl

**Keywords:** epoxy composites, glass fabrics, aramid fabrics, carbon fabrics, basalt fabrics, flax fabrics

## Abstract

The production of hybrid layered composites allows comprehensive modification of their properties and adaptation to the final expectations. Different methods, such as hand lay-up, vacuum bagging, and resin infusion were applied to manufacture the hybrid composites. In turn, fabrics used for manufacturing composites were made of glass (G), aramid (A), carbon (C), basalt (B), and flax (F) fibers. Flexural, puncture impact behavior, and cone calorimetry tests were applied to establish the effect of the manufacturing method and the fabrics layout on the mechanical and fire behavior of epoxy-based laminates. The lowest flammability and smoke emission were noted for composites made by vacuum bagging (approximately 40% lower values of total smoke release compared with composites made by the hand lay-up method). It was demonstrated that multi-layer hybrid composites made by vacuum bagging might enhance the fire safety levels and simultaneously maintain high mechanical properties designed for, e.g., the railway and automotive industries.

## 1. Introduction

Composites were created due to endeavoring to obtain materials with high strength and simultaneously low density. Widely used in industry, thermoset composites (mainly based on unsaturated polyester and epoxy resins) reinforced with long fibers in the form of roving bundles or fabrics, thanks to their specific strength, become competitive to metals in many applications [1]. Depending on the type of fibrous reinforcement, layered and hybrid composites can be gained [2,3]. In recent years, much research has been devoted to hybrid composites, where two or more reinforcements are used to modify a polymer matrix. The main goal is to overcome the drawbacks of the single reinforcement by adding another type of fiber [4,5]. Furthermore, the cost of the one type of fiber may be high, and a combination of different fibers allows obtaining the desired properties at a lower price. For example, hybridization with basalt fibers (BF) of epoxy–carbon fibers (CF) composites may improve their damping properties [6], and application of an external CF layer into glass fiber (GF)-reinforced epoxy bars allows achieving higher resistance to shear strength after accelerated aging in water [7]. Nevertheless, improvement in mechanical properties, such as increasing the low-velocity impact strength, will inevitably involve the need to introduce additional layers in the laminate [8]. In turn, the choice of manufacturing technology depends on the shape, dimensions, and final use of the composites.

In the literature, discussing the potential benefits resulting from the use of selected methods can be found. Kim et al. [9] compared the effect of the manufacturing technique on the properties of hybrid composites with a matrix of vinyl ester resin cross-linked with methyl ethyl ketone peroxide (MEKP) and reinforced with glass fiber. Composites made by the hand lay-up method and resin infusion were subjected to static tensile and compression tests. It was shown that the materials produced by the resin infusion method were characterized by almost two times higher strength parameters at break than the samples shaped with the use of manual one, with the simultaneous increase in the elongation at break values by about 30%. In the case of the compression test, the composite materials showed comparable values of compressive strength (approximately 350 MPa), while the maximum stress value in the case of composites prepared by the resin infusion was achieved at 65% of the deformation value than with materials formed by the manual method. The study by Astrom [10] compared the strength properties of thin-walled composites made by the vacuum bagging and resin infusion technique. One-filler and hybrid epoxy composites reinforced with carbon and glass fibers were used for the research. The authors showed that the composites made by the infusion technique were characterized by more favorable mechanical properties than those formed by the vacuum method. On the other hand, Najfi et al. [11] analyzed the influence of the production technique on the properties of hybrid sandwich composites with a polyester resin as a matrix. Composites reinforced with glass and aramid fibers with a balsa wood core were subjected to a compressive strength test. The resin infusion method allows obtaining materials with higher compressive strength than hand-formed products, which is caused by a greater degree of supersaturation of the inner reinforcing layer. Rydarowski and Koziol [12] compared the properties, quality, and repeatability of composites made using a polyester resin matrix reinforced with fabrics and fiberglass mats manufactured by hand lay-up and resin infusion methods. Materials produced using the resin infusion method reinforced with glass mats were characterized by twice the volume of fibers compared with those shaped by manual ones. However, in the case of using glass fabrics as a filler, the volumetric content of the resin in the composites was similar. The strength properties, including flexural strength and impact strength, of the composites produced by both methods were comparable for fabric-reinforced resins, while the use of mats was associated with obtaining higher strength properties by composites made by resin infusion. At the same time, it should be emphasized that the scatter of the mechanical strength results for hand-formed composites was twice as large. Composites prepared by resin infusion were also characterized by greater repeatability and uniformity of dimensions than those formed by the manual method.

While the hand lay-up method is the most commonly used technique of laminates forming, which allows for the production of finished products almost without the use of expensive tooling, numerous studies have shown that it is challenging to obtain high-quality products with a limited number of structural defects and a high concentration of fillers with this technology. Therefore, different low-cost manufacturing methods are frequently used in industrial practice, allowing for a higher degree of fiber supersaturation and minimizing the risk of voids and pores. Among technologies that would enable thin-walled products to be reinforced with continuous fibers, large surfaces, and complicated shapes, vacuum bagging (VB) and vacuum-assisted resin infusion (VARIM) [9,11,12] should be mentioned. At the same time, pultrusion is one of the more developed technologies allowing outstanding mechanical performance [3,5,7,13,14]. While pultrusion technology also allows for the formation of continuous products, including plates, with excellent mechanical properties and almost any possibility of shaping the order of the fibers, they are intended instead for unidirectional products production. Therefore, the rest of the study focuses on comparing the properties of laminates produced by VB and VARIM techniques compared with hand lay-up.

Many papers describe the effects of the configuration of multi-layer laminates manufactured with various reinforcement materials. However, it should be emphasized that usually these works relate to systems containing two or three types of fabrics, focused on using one with natural origin (e.g., kenaf fiber) or high performance (e.g., carbon fiber) [15,16,17,18]. In most cases, the impact of ratio and mutual layering in the composite was the most crucial issue. At the same time, considering the flammability of hybrid composites, systems containing up to two types of fibers are investigated, with the simultaneous introduction of a flame retardant or inorganic particle-shaped fillers [19,20,21,22]. Still, no comprehensive studies have been reported on evaluating multi-material laminates produced by various manufacturing techniques, considering the analysis of mechanical and flammability properties related to the structure and quality of final products.

The presented studies of hybrid composites are evaluated by low-impact drop and three-point bending tests compared with the results of flammability analysis using the cone calorimetry method. Numerous studies have shown that the simultaneous introduction of several appropriately selected types of fabrics with different properties (e.g., nylon–basalt [23], glass–kenaf [24]) allows increasing low-impact strength by obtaining a layer-by-layer failure mode, translating into a much larger energy absorption [25,26]. Mechanical performance, considered in terms of flexural strength of hybrid composites, depends on the individual fibers’ strength and stiffness. Thus, the introduction of higher stiffness fillers to the system will, each time, be associated with the noticed changes in the strength of the product (e.g., the introduction of carbon fibers to the glass–epoxy composite) [27]. However, bending strength deterioration may occur at the expense of improving other properties, such as increasing the resistance to vibration damping (carbon–flax) [28]. Taking into account the third of the discussed characteristics, i.e., lower flammability, the introduction of reinforcing layers with higher temperature stability, such as basalt, glass, or carbon, will allow increasing the thermal resistance of the final components, in contrast with products based on natural fibers [29,30].

In this study, five fabrics were used to manufacture composites by the hand lay-up, vacuum bagging, and resin infusion methods. Research methods, including flexural tests, puncture impact behavior, and cone calorimetry tests, were applied to establish the impact of the manufacturing method and the fabrics layout in laminate structure on the mechanical and fire behavior of epoxy-based composites. Moreover, the properties’ evaluation was related to the microstructure analysis carried out before and after the cone calorimetry tests. The investigation constituted a continuation of the results presented in our previous work [31].

## 2. Experimental

### 2.1. Materials and Laminates Fabrication

The epoxy resin RenLam LY 113 (viscosity 580 mPa∙s at 25 °C, density 1.16 g/cm^3^) and a curing agent Ren HY 97-1 (viscosity 20 mPa∙s at 25 °C, density 0.95 g/cm^3^), supplied by Huntsman Advanced Materials GmbH (Basel, Switzerland), were used as the polymer matrix. Processing of the composites was performed using resin to curing agent ratio 100:30. The following fabrics were used as a reinforcement: a two-way (+45/−45°) sewn glass X-E (G) made of E-glass and with a grammage of 444 g/m^2^ from Saertex GmbH & Co. KG (Saerbeck, Germany), aramid fabric (A) made of Tex 240 fibers with a 2 × 2 twill weave and a weight of 300 g/m^2^ delivered by P.P.H.U. SURFPOL (Rawa Mazowiecka, Poland), a two-way (+45/−45°) sewn carbon X-C (C) with a grammage of 406 g/m^2^ from Saertex GmbH & Co. KG (Saerbeck, Germany), a two-way (+45/−45°) sewn basalt fabric BAS BI 450 (B) with a grammage of 464 g/m^2^ from Basaltex NV (Wevelgem, Belgium), a flax fabric (F) with a 2 × 2 twill weave and a grammage of 500 g/m^2^ made of Tex 400 fibers manufactured by Safilin (Sailly-Sur-La-Lys, France).

The fabrics with 330 × 330 mm^2^ were placed one by one on an ultra-high molecular weight polyethylene plate coated with Teflon foil. At the same time, the epoxy resin components were mixed by a proLAB 075 mechanical stirrer with a rotational speed of 2000 rpm for 3 min and under subatmospheric pressure. In the hand lay-up method, an epoxy resin was uniformly dispersed by a roller. Additionally, for the vacuum bagging method, they were next wrapped in a peel-ply vacuum blanket and vacuum bag before vacuuming was started to aspirate the excessive resin. In the resin infusion, after the fabrics were covered by a peel-ply, net, and vacuum blanket, a vacuum was applied from one end of the bag, while the resin was fed from the other to wet the fabrics entirely. Samples were kept under a vacuum until the curing process was completely finished. After the forming process and based on our previous work [31], the laminates were cured at room temperature for 72 h and post-cured at 70 °C for the next 3 h. Last, the composites were cut precisely for conducting the measurements. Each composite consisted of five two-ply layers and was produced in two different arrangements: glass/aramid/carbon/basalt/flax (GACBF) or aramid/glass/carbon/basalt/flax (AGCBF), and Table 1 summarizes the configurations of produced specimens. The fabrics layout was based on preliminary research presented in [32]. The results of tests carried out using the cone calorimeter were the most significant in selecting the sequence of fabrics.

### 2.2. Methods

Cross sections of the obtained composites were examined using an ultra-high resolution scanning electron microscope SU8010 (Hitachi, Japan). The samples were sputtered with gold, coated by a Quorum Technologies Q150T ES sputter coater. Observations were conducted at an accelerating voltage of 10 kV, at magnification × 100 and the biggest possible working distance, usually WD > 30 mm, to maximize field depth and minimize image distortion. Each sample was oriented the same and observed from top to bottom. Images were taken at a constant stage shift to ensure a sufficient area overlap. For each sample, partially overlapping images were stitched together using Grid/Collection Stitching plugins available in the open-source image processing package Fiji suite [33]. The number of images depended on the composite height, usually 8–10.

The flexural tests were performed using a universal machine, an ElectroPuls E3000 from Instron (Norwood, MA, USA), following ISO 14125 standard. The analyses were carried out with 6 × 15 mm^2^ samples at a 7 mm/min crosshead speed and a support spacing of 110 mm. The experiments were conducted at ambient temperature for at least three samples from each series.

The low-speed impact test was conducted on an INSTRON Dynatup 9250 HV Impact Tower test machine (Norwood, MA, USA) according to ASTM D7136/D7136M. The total mass of drop weight framework with a hemispherical striker equaled 5830 g. The hemispherical striker with a diameter of 25.4 mm was connected with a force transducer of 16 kN capacity to estimate the impact force during the test. Three composites of each series with 100 × 100 × 5 mm^3^ dimensions were clamped between two steel plates with a circular-cut hole 40 mm in diameter. The falling height between the striker and the top surface of the specimen was 7.0 m. The impact energy could be calculated using a mass of drop weight framework and falling height from the potential energy equation. The calculated impact energy was chosen for the failure of all samples without a rebound. In the present work, the impact energy of 400 J was calculated. Absorbed energy, velocity, and deflection were computed during the experiment using the dedicated software.

A cone calorimeter from Fire Testing Technology Limited (East Grinstead, UK) was used to identify the burning behavior under forced-flaming conditions. Samples with the dimensions of 100 × 100 × 5 mm^3^ were placed in an aluminum foil tray and exposed horizontally to an external heat flux of 35 kW/m^2^ located 25 mm away from the cone heater. The test was conducted following the ISO 5660 standard. Three samples from each series were tested, and more measurement was carried out if the key result (pHRR) varied by more than 10%. After the tests, the samples were photographed using a digital camera, EOS 400 D from Canon Inc. (Tokyo, Japan).

Samples after cone calorimetry were also investigated by SEM; however, did not require sputter coating. Due to their high porosity and brittleness, composite chars were fixed on the table using LEIT-C Conductive Carbon Cement. Both top surface and bottom chars were observed and analyzed, as well as each fiber type. Point elemental analysis was carried out using the Thermo Scientific NORAN System 7 equipped with an electrically cooled Silicon Drift Detector EDS detector (Thermo Scientific UltraDry, Waltham, MA, USA) to investigate the chemical difference in top surface area and bottom surface area of the chars.

## 3. Results and Discussion

### 3.1. Surface Morphology Analysis

Microscopic images of cross sections, presented in Figure 1, displayed a morphology with well-penetrated fibers by the resin. Differences in appearance and various directions of the arrangement of fabrics allowed for the separation of individual layers. The images show bundles of stiff fibers from sewn fabrics (glass, basalt, carbon), split and sometimes taking different fibers from woven fabrics (aramid, linen). Observable bundles of smooth glass, basalt, or carbon fibers, as well as long and tangled aramid or linen fibers, indicate limited adhesion of resin and reinforcement. Based on the analysis of images of H-GACBF, V-GACBF, and I-GACBF, it was observed that the lowest number of voids occurred for composites made by resin infusion, then hand lay-up and vacuum bagging. This effect applied especially to the outer layer, suggesting that the vacuum was used for too long. Photographs showing the saturation of the top layer of samples are included in the Appendix A (Appendix A). The presence of voids is the main disadvantage of the hand lay-up method, similar to the matrix’s unfavorable ratio to the reinforcement.

### 3.2. Mechanical Properties Evaluation

The effect of the type of fiber reinforcement and the method of composites production on the mechanical properties were defined by flexural and low-speed impact tests. Figure 2 depicts the flexural strength (σ_f_) and Young’s modulus (Et) values determined from flexural measurements.

In the case of composites made by the hand lay-up method, H-AGCBF was characterized by higher flexural strength, and the value was slightly above 200 MPa. For composites manufactured by the vacuum bag method, about 200 MPa was also obtained for the series in which the aramid fabric was the most outer layer. During a static three-point bending test, the material layer opposite to the middle point is subjected to a dominant state of stress in the tensile mode [34]. In the case of bending, the final flexural strength of the laminates depends on the crack propagation mechanism resulting from interlaminar adhesion and the tensile strength of the external layer [35]. Epoxy composites with aramid fibers are characterized by higher tensile strength than glass fibers [32,36]; therefore, the obtained results may be related to this effect. Similarly, both composites made by resin infusion were characterized by flexural strength above 150 MPa, and the highest value (190 MPa) was obtained for the AGCBF. The effect of the arrangement of fabrics during the study of hybrid composites was described by Abd El-Baky et al. [37] and Sarasini et al. [38]. These results also agree with the data presented by Guo et al. [13]. In the case of layered materials, when the critical values of bending stress are exceeded, interface debonding takes place, and subsequently redistributes stress to the next layer, up to the achievement of fracture of the last layer [13]. In a comparison of all the methods, it was observed that slightly higher results were obtained for composites made by the hand lay-up method, followed by vacuum bagging and infusion. This is a somewhat surprising conclusion and probably resulted from insufficient saturation of the first layer, which was considered when discussing the microstructure of composites (Figure 1 and Appendix A). Similar relationships were observed for the stiffness shown by the Young’s modulus value in Figure 2b. This is because both flexural strength and stiffness are controlled mainly by the outer layers [39,40,41]. Young’s modulus values of almost all composites, except H-GACBF, were above 9 GPa, and the highest results were obtained for H-AGCBF and I-AGCBF. Analysis of the results showed that the fabrics layout had a more significant influence on the results than the production method.

The evaluation of the low-speed impact test was realized by assessment of the damage of composites made by the impactor through visual investigation. In Figure 3, damage on the laminates’ top (impacted side) and bottom surfaces can be found. The damaged areas of all samples were marked and may be characterized with a good agreement with Barley Visible Impact Damage (BVID) modes [42]. According to ASTM D7136M–12 standard, all composites revealed “Combined Large Cracks with Fiber Breakage, Indentation/Puncture” damage mode. In the case of GACBF composites, matrix cracking, fiber breakage, and delamination around the impact point were observed [43]. The failure mode is similar to well-known damage modes in laminates, which depend on the impact energy and impactor shapes [31]. The most significant visible delamination around the impact point, reaching approximately 44 mm, was noted for GACBF manufactured by the hand lay-up method. This result contradicts that observed for glass-reinforced polyester laminates [44], where vacuum infusion-made samples showed a higher ability to delamination than the hand lay-up method. The difference may result from a decidedly simple damage mechanism for composites made of one type of reinforcement. In turn, the damage size for AGCBF samples was independent of the manufactured method and reached about 28 mm. Considering the results presented by Atas et al. [45], the observed comparable damage formation of laminates is justified due to the high impact energy (400 J).

The use of a hybrid arrangement of fibers in a laminate allows for many times higher impact resistance values than in the case of single-type fiber composites [46]. Figure 4 was plotted to illustrate the impact energy of analysis hybrid composites. It can be observed that the maximum value of absorbed energy was presented by V-GACBF and I-AGCBF. In turn, the lowest energy needed to fracture the laminate in a low-speed impact test was revealed by the H-GACBF composite series. The results evidence that a higher resistance is offered by composites made by the vacuum methods. The fabrics layout was also significant. In most cases, higher values were obtained for composites with aramid as the outermost layer. The only exception was the V-AGCBF, for which the largest scatter of values was also noted. In Appendix A, presented in Appendix A, the course of force vs. deflection of the impactor during weight drop measurements present a multi-stage cracking course characterized by the samples with the highest values of absorbed impact energy. Thus, reducing the amount of resin in the composite allowed for a more effective load transfer through all the applied reinforcing layers of the filler. Variations between series may also be related to the different in-plane stiffness of the composites [47]. All samples showed an open-shaped character of force–deflection curves. In turn, the energy–deflection curves exhibit all three characteristic failure modes. The first step consists of the indentation, dent formation, matrix breakage, and delamination, while the second step of failure refers to fiber fracture at in-plane and through-thickness direction (an energy increase). In the last region, energy vs. deflection lines show constant energy, which is related to the reception of the total impact energy, resulting in a complete loss of the composite structure [48].

In most studies, there is a slight correlation between the applied laminate production technique and the impact strength measured using various methods, including Izod, Charpy, or falling weight test [12,45,49]. However, it should be emphasized that in most of the works lower than in the considered case impact energies (up to 100 J) were used. To conclude, it should be emphasized that despite differences in the measured values, all composites were characterized by exceptionally high impact resistance and the ability to suppress the impact energy. The obtained measurement data are consistent with the results presented by Jang et al. [50]. The authors stated that it is unclear which type of fibers (glass or polymeric) should be in the layer facing the impactor. However, the use of synthetic fibers, with a high capability to be stretched to a large extent during impact load, will benefit the impact energy distribution. In turn, highly durable polymer fibers in the interlayer will positively influence the formation of delaminations on a larger area, causing energy dissipation.

As demonstrated, the mechanical properties depend on the fabric configuration, as well as the measurement technique used [51]. Considering the importance attached to the need for high impact resistance of laminates and the resulting ability to enhance post-impact performance [26], increasing impact strength will be crucial for applications in the railway or automotive industry [52]. Based on the obtained test results, it is recommended to manufacture products using vacuum-assisted methods, the use of which results in higher impact properties of the laminates than use of the hand lay-up technique.

### 3.3. Burning Behavior

A cone calorimeter was applied to assess the fire performance of materials under conditions that simulate developing fires. Several critical parameters, including peak heat release rate (pHRR), time of ignition (TTI), total heat release (THR), total smoke released (TSR), specific extinction area (SEA), total smoke release (TSR), and the maximum average rate of heat emission (MARHE), are summarized in Table 2. Figure 5 plots the HRR curves, while the char residues of the hybrid composites are shown in Figure 6.

Figure 5 reveals that all HRR curves contain two peaks and a plateau-like behavior between them or sometimes three peaks. All composites’ curves present pHRR at the end of the burning before rapidly decreasing. Moreover, the pHRR values were similar, except for V-AGCBF, and did not significantly depend on the production method used.

The V-AGCBF was characterized by the longest time after which the flame appeared on the entire surface and was maintained until it was extinguished, reaching 189 s. In turn, the lowest values were achieved for composites prepared by the hand lay-up process, which suggests that TTI was conditioned by the manufacturing method. Moreover, only in the case of the hand lay-up method did the material with glass fabric as the outermost layer, known to be fire material resistant, have a longer time to ignition than aramid. The amount of resin in the outer layer had a significant impact on the obtained results, which, as revealed by the microstructure analysis, was lower in the case of composites made using vacuum bagging. The heat release rate, as the critical parameter among the fire properties, ranged from 219 to 636 kW/m^2^, and the lowest values were also obtained for composites V-AGCBF. Since the reinforcements and their order were similar, it confirms the manufactured method’s crucial influence and, thereby, the resin’s content as the most flammable component. Similarly, the lowest values of MARHE, used to estimate the hazard of fire spread [53], were obtained for V-GACBF. Contrary to the pHRR, where the meaningful differences between the values (confirmed by the standard deviation) make it difficult to observe any trend, MARHE revealed that the lowest values were recorded for laminates made by the vacuum bagging, then resin infusion and hand lay-up method. Notable differences in the THR results obtained for various laminates manufacturing techniques, with a predominance of laminates made using the vacuum bagging method, were also noted. Total heat release is considered as a measure of the fire load, suggesting incomplete combustion by reducing combustion efficiency and/or the char formation [54]. However, the photographs of the samples after fire tests, shown in Figure 6, disclose that residues consist mainly of fabrics. It suggests that the lower amount of the resin in the case of samples made with the vacuum bagging method caused a reduction in the emission of volatile decomposition products into the combustion zone.

Photographs of the samples after the cone calorimeter test (Figure 6) reveal the appearance of the sequentially stacked fabrics used to prepare the composites, while the char residue on the surface of the samples is barely visible. Composites made by the vacuum bagging method resemble the others, but unlike the composites made by the hand lay-up method, their outer layers were less damaged. On the other hand, laminates prepared using the resin infusion method were characterized by a degree of sample destruction similar to that observed for composites made by the hand lay-up method. The appearance of all samples reflects the intensity of the burning process.

Additionally, the unprocessed fibers and residues were subjected to microstructure analysis using SEM to estimate the destruction of laminates after the fire tests. In the left column of Figure 7, each fabric’s original appearance is presented, while the subsequent layers of a composite V-AGCBF after the cone calorimetry measurements are compiled in the right column. The most damaged is the outer layer, consisting of two plies of aramid fabric. Images at higher magnifications revealed that the fibers had started to melt and sometimes stick together under the impact of high temperature (Figure 8). Below it, there is less damaged glass fabric and next carbon fabric. As shown in Figure 8, some changes in carbon fibers surface can also be observed. The following layer was basalt fabric, which looks similar to the original ones. In turn, the fibers of the last fabric are stuck together by the polymer matrix, suggesting that the reinforcements above prevented the resin from burning out completely.

Similar to the previously discussed parameters, the lowest values for smoke emission were recorded for V-AGCBF (1873 m^2^/m^2^) and V-GACBF (2047 m^2^/m^2^). In turn, the highest result (approximately 40% higher) was noted for composites made by the hand lay-up method, which had the matrix’s most unfavorable ratio to reinforcement. As can be seen in Figure 9, composites demonstrated an amount of smoke emission according to the scheme below:vacuum bagging < resin infusion < hand lay-up method

Moreover, a slight delay in emissions for laminates manufactured by the vacuum bagging method can also be observed. Apart from the production method, the arrangement of fabrics had some influence, and lower TSR was recorded for laminates with aramid fabrics as the outer layer.

## 4. Conclusions

This work provides a comparative investigation of the effects of reinforcement order and manufacturing method on mechanical properties and burning behavior of EP-laminates. Aramid, glass, basalt, carbon, and flax fiber-reinforced composites with two configurations were fabricated using vacuum bagging, resin infusion, and hand lay-up method.

The used methods allowed the preparation of well-saturated composites, however, each with a different amount of resin. The flexural tests suggested that fabrics layout and the choice of the outer layer may have had the main impact on the result, while the low-speed impact and cone calorimetry tests revealed only their minor influence. The matrix’s unfavorable ratio to reinforcement in the case of hybrid composites prepared by the hand lay-up method caused a reduction in absorbed energy and an increase in the intensity of burning. In turn, the lowest flammability and smoke emission, while maintaining good mechanical properties, were noted for composites made by vacuum bagging.

It was demonstrated that multi-layer hybrid composites could achieve improvements in lightweight design and simultaneously maintain high fire safety levels in, e.g., the railway and automotive industries.

## Figures and Tables

**Figure 1 materials-15-00301-f001:**
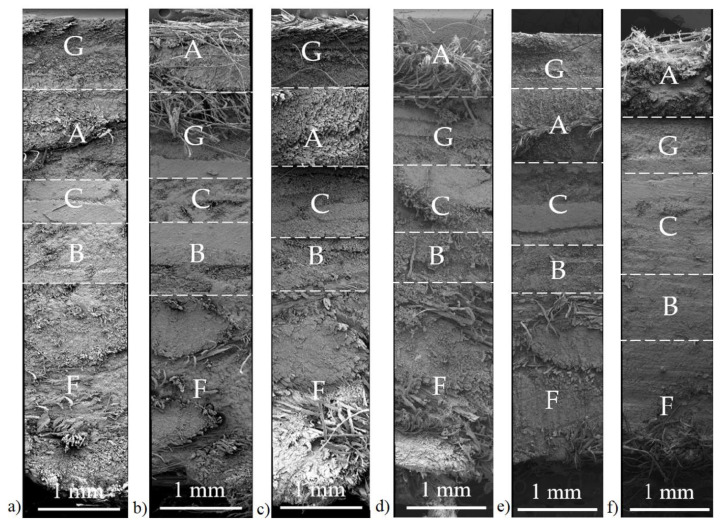
SEM images of cross sections of hybrid composites: H-GACBF (**a**), H-AGCBF (**b**), V-GACBF (**c**), V-AGCBF (**d**), I-GACBF (**e**), I-AGCBF (**f**).

**Figure 2 materials-15-00301-f002:**
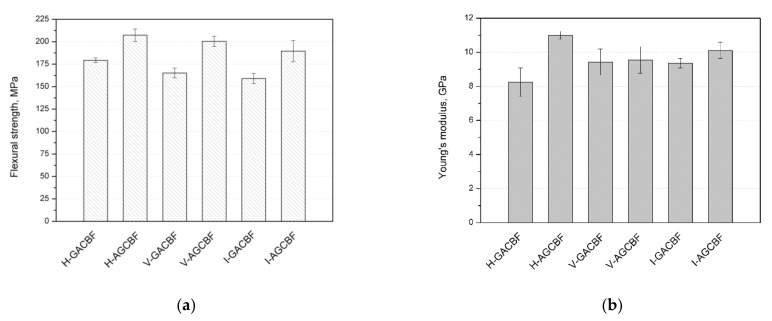
Flexural strength (**a**) and Young’s Modulus (**b**) results of hybrid composites obtained from the flexural measurements.

**Figure 3 materials-15-00301-f003:**
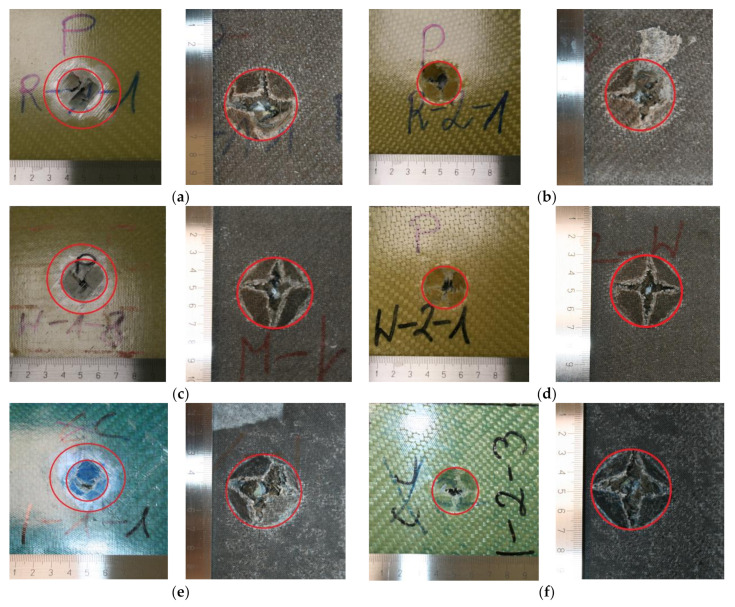
Photographs of internal and external surfaces of hybrid composites H-GACBF (**a**), H-AGCBF (**b**), V-GACBF (**c**), V-AGCBF (**d**), I-GACBF (**e**), I-AGCBF (**f**) after impact test.

**Figure 4 materials-15-00301-f004:**
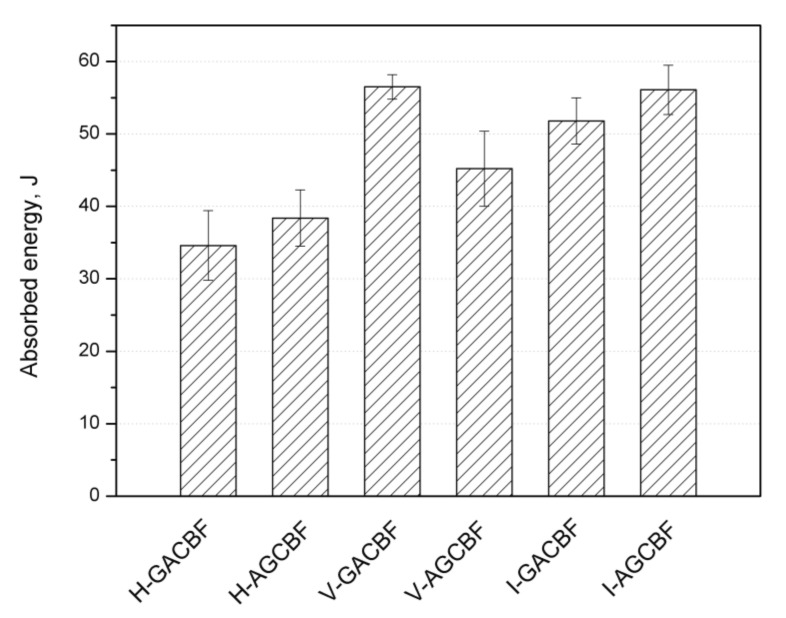
The absorbed energy of EP hybrid composites measured by drop test.

**Figure 5 materials-15-00301-f005:**
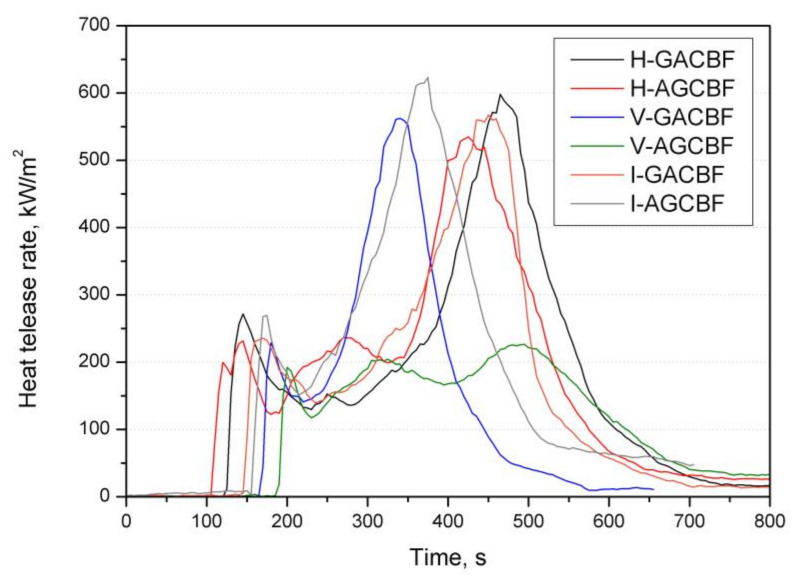
Representative curves of the heat release rate of the EP-based composites.

**Figure 6 materials-15-00301-f006:**
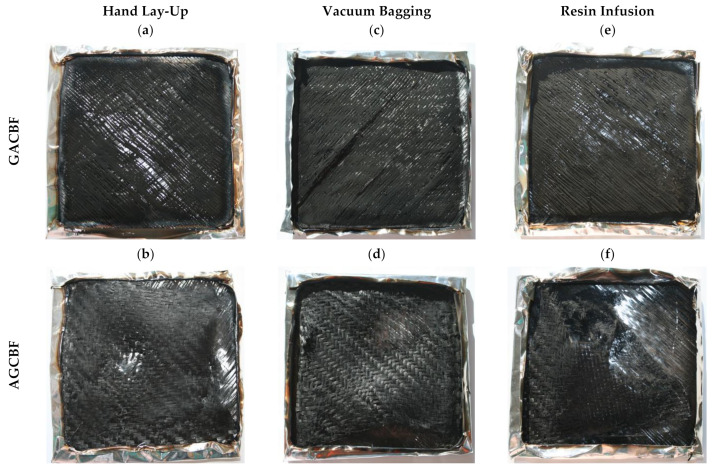
Photographs of GCABF (**a**), CGABF (**b**), CAGBF (**c**), MAGCBF (**d**), VGACBF (**e**), MGACBSF (**f**) after cone calorimetry tests.

**Figure 7 materials-15-00301-f007:**
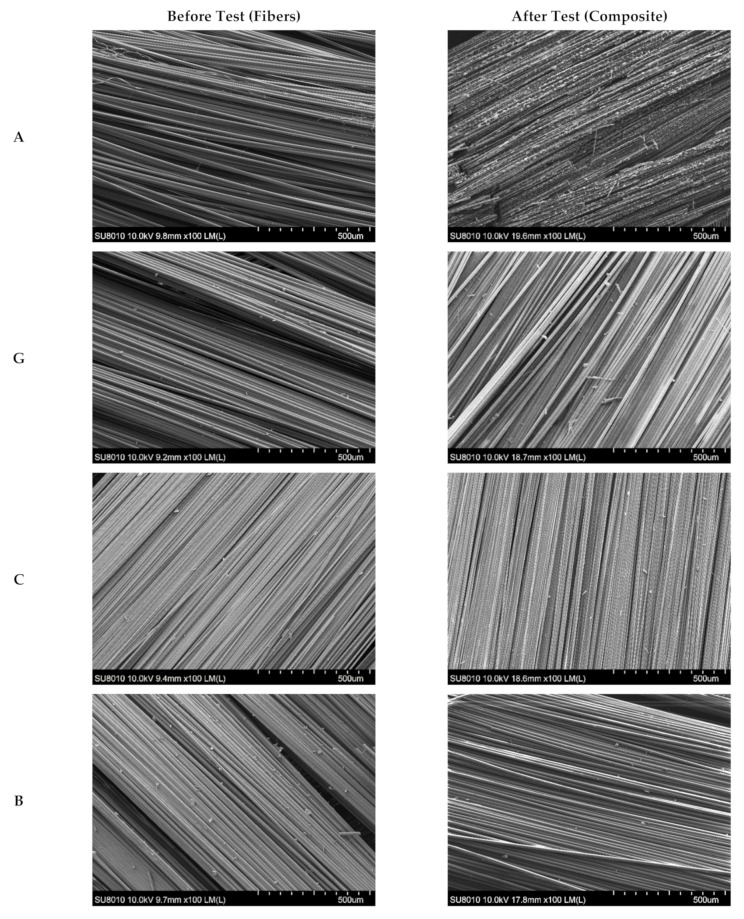
SEM images of A, G, C, B, and F fabrics before burning tests (**left** column) and V-AGCBF laminates after the burning test (**right** column).

**Figure 8 materials-15-00301-f008:**
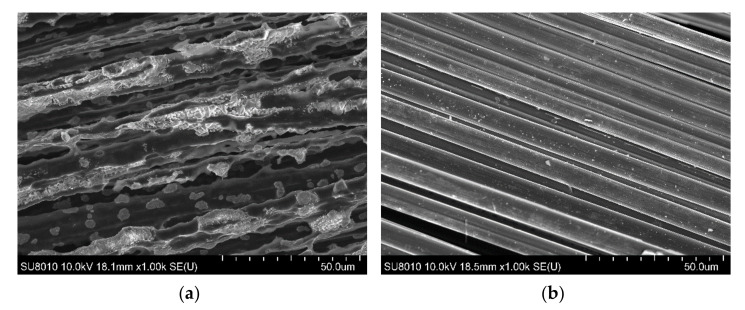
SEM images of A (**a**) and C (**b**) fabrics after fire tests.

**Figure 9 materials-15-00301-f009:**
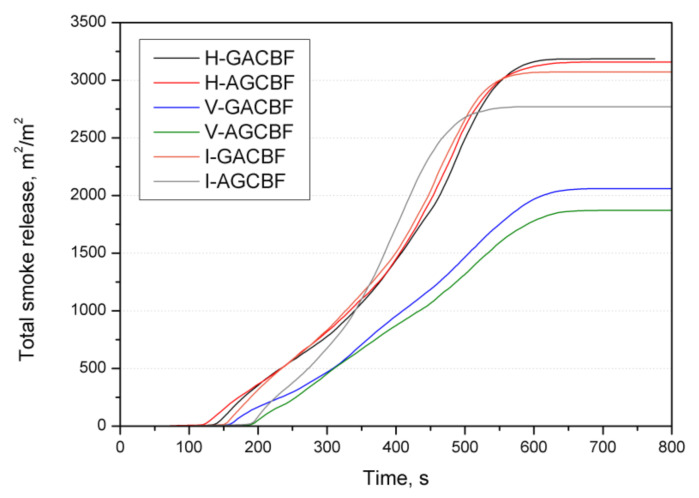
Representative curves of the total smoke release of the EP-based composites.

**Table 1 materials-15-00301-t001:** Sample assignment including laminate stacking sequences and manufacturing method.

Samples	G	C	A	B	F	Manufacturing Method
H-GACBF	2a	2b	2c	2d	2e	hand lay-up
H-AGCBF	2b	2a	2c	2d	2e	hand lay-up
V-GACBF	2a	2b	2c	2d	2e	vacuum bagging
V-AGCBF	2b	2a	2c	2d	2e	vacuum bagging
I-GACBF	2a	2b	2c	2d	2e	resin infusion
I-AGCBF	2b	2a	2c	2d	2e	resin infusion

2 is the number of fabrics, while a–e is the fabrics layout.

**Table 2 materials-15-00301-t002:** Cone calorimeter results of hybrid composites.

Samples	TTI, s	pHRR, kW/m^2^	MARHE, kW/m^2^	THR, MJ/m^2^	TSR, m^2^/m^2^
H-GACBF	129 (7)	544 (183)	200 (28)	131 (3)	3211 (54)
H-AGCBF	118 (10)	595 (86)	226 (7)	136 (4)	3121 (51)
V-GACBF	164 (7)	562 (1)	177 (0)	81 (4)	2047 (240)
V-AGCBF	189 (4)	219 (10)	116 (8)	83 (3)	1873 (1)
I-GACBF	159 (11)	496 (160)	192 (24)	126 (10)	3114 (166)
I-AGCBF	174 (17)	636 (30)	211 (13)	115 (5)	2716 (147)

The values in parentheses are the standard deviations.

## Data Availability

All Data contained within the article.

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
