# Peer review of "The Effect of Manufacture Process on Mechanical Properties and Burning Behavior of Epoxy-Based Hybrid Composites"

_materials, 2022, doi:10.3390/ma15010301_

Round 1

Reviewer 1 Report

A Hand lay-up, vacuum bagging, and resin infusion methods have been used to produce five fabrics with similar grammage. Some experimental analysis (such as flexural, puncture and cone calorimetry tests) are performed to establish the impact of the manufacturing method and the fabrics layout in laminate structure on the mechanical and fire behavior of epoxy-based composites. It is concluded that multi-layer hybrid composites could improve the light- weight design and simultaneously maintain high fire safety levels. The paper is well written and the results are interesting, so it can be accepted for publication.

Author Response

The authors would like to thank the Reviewer for the effort to revise our article and for the positive and edifying opinion.

Yours faithfully,

Kamila Salasinska

Reviewer 2 Report

The mamuscript reports an accurate comparative investigation of composite systems characterised by a different stacking sequence and hybridization of reinforcing fiber in respect to the fire behaviour and flexural/impact performance. The paper has some technical merit but issues arise and they are asked to be revised accordingly.

Please consider the following comments:

a) the abstract should provide insights regarding the achieved results

b) the definition of the stacking sequence has not been specified. PLease add the reason of the determined choice and the criteria for the implementation of the type of hybridization

c) please specify the matrix content as in different sections it was reported that the composites "with different amount of resin"

d) the procedure of impact test could be better reported and related to a standard. Thi sis necessary as from the reported picture the indentation appear very close to the sample edge and this condition could be invalidating the results

e) why the work was conducted directly assuming the hybridization layout? what considering a plain composite and make the corresponding comparison with the hybrid configuration?

I will be willing to review this paper after minor revision before publicaiton.

Author Response

We highly appreciate all the comments and find them very useful. We agree with the recommendations that the manuscript should be improved, so efforts have been made to correct the article according to the comments. Below we enclose the replies to the comments and recommendations made by the Reviewer. All of the changes in the text are highlighted using colour.

  1. The abstract should provide insights regarding the achieved results. 

We thank the Reviewer for comment. The abstract has been revised according to comments from all Reviewers.

  1. The definition of the stacking sequence has not been specified. Please add the reason of the determined choice and the criteria for the implementation of the type of hybridization.

The results presented in the article were carried out as part of the first stage of the 3-years project, and the aim was to select the manufacturing method. Based on the literature data and team members' experience, several different reinforcements in the form of fabrics and powder fillers were selected [Sałasińska, K.et al. Experimental Investigation of the Mechanical Properties and Fire Behavior of Epoxy Composites Reinforced by Fabrics and Powder Fillers. Processes 2021, 9, 738]. The task of the next stage will be the selection of the composition and its optimization determined by the application. The stacking sequence was based on our preliminary research. The article [Salasinska, K.; Barczewski, M.; Aniśko, J.; Hejna, A.; Celiński, M. Comparative Study of the Reinforcement Type Effect on the Thermomechanical Properties and Burning of Epoxy-Based Composites. J. Compos. Sci. 2021, 5, 89] presents the results of mechanical and burning tests for composites with the same fabrics but one type of reinforcement. The results of the tests carried out using the cone calorimeter were decisive in selecting the sequence of fabrics. The compositions presented in this article are only a fragment of the research to show established relationships and constitute an introduction to the following works.

We thank the Reviewer for the valuable comment. We agree with the recommendation and carefully improved that inaccuracy in the manuscript.

  1. Please specify the matrix content as in different sections it was reported that the composites "with different amount of resin". 

We thank the Reviewer for the valuable comment; however, in the case of composites produced with natural fibres, or hybrid composites with different fibres, it is difficult to specify the actual amount of resin. Bearing in mind that the thermal degradation of some of the fillers used coincides with the degradation of the epoxy resin, it will be difficult to obtain quantitative information on the proportion by thermogravimetric analysis. The authors made an attempt to determine the proportion of resin based on the results of the density determined by the hydrostatic method, and a the results are presented below. Whereas, as far as the total share of resin of samples made with the hand lay-up method could be estimated based on the study of the density, for the vacuum methods, due to its partial removal of resin during the manufacturing, will not be truthfully and appropriate.

Samples

G

C

A

B

F

Manufacturing Method

Density, g/cm3

H-GACBF

2a

2b

2c

2d

2e

hand lay-up

1.329 ± 0.054

H-AGCBF

2b

2a

2c

2d

2e

hand lay-up

1.357 ± 0.025

V-GACBF

2a

2b

2c

2d

2e

vacuum bagging

1.341 ± 0.023

V-AGCBF

2b

2a

2c

2d

2e

vacuum bagging

1.358 ± 0.043

I-GACBF

2a

2b

2c

2d

2e

resin infusion

1.394 ± 0.001

I-AGCBF

2b

2a

2c

2d

2e

resin infusion

1.266 ± 0.091

2 is a number of fabrics, while a–e is the fabrics layout; EP density 1.115 ± 0.001

  1. The procedure of impact test could be better reported and related to a standard. Thi sis necessary as from the reported picture the indentation appear very close to the sample edge and this condition could be invalidating the results.

We thank the Reviewer for the remark. The description of the procedure of the impact test has been changed accordingly. The measurements were conducted according to ASTM D7136/D7136M Standard Test Method for Measuring the Damage Resistance of a Fiber-Reinforced Polymer Matrix Composite to a Drop-Weight Impact Event. In the photos, a ruler is placed on the samples to better show the area of destruction, not the dimensions of samples. The dimensions of each sample were 100 × 100 × 5 mm, as can be observed in the photograph below.

  1. Why the work was conducted directly assuming the hybridization layout? what considering a plain composite and make the corresponding comparison with the hybrid configuration?

As we mention above, the hybridization layout was made based on preliminary work. The article [Salasinska, K.; Barczewski, M.; Aniśko, J.; Hejna, A.; Celiński, M. Comparative Study of the Reinforcement Type Effect on the Thermomechanical Properties and Burning of Epoxy-Based Composites. J. Compos. Sci. 2021, 5, 89] presents the results of mechanical and burning tests for composites with the same fabrics but one type of reinforcement. Thanking the Reviewer for the remark, we would like to say that the explanation has been introduced to the article, and the results presented in the article were related to the preliminary research.

We have also corrected quite a number of other minor errors which we noticed while working on the text, and we believe that our manuscript in the present form can be published in the Journal.

Yours faithfully,

Kamila Salasinska,

Mateusz Barczewski

Reviewer 3 Report

The authors propose a comparison of different hybrid composite layouts and manufacturing methods.

Overall, this is an good article for the objectives proposed but some ideas need to be clarified before the article can be published.

Abstract, line 23 - It is said that the different fabrics of the different fibers used in this study are of "similar grammage".

For example the grammage of the: aramid fabric (A) is 300 g/m2 and flax fabric (F) is 500 g/m2. the difference is obvious for the authors to be able to assert this. This difference is also evident in Fig. 1 (SEM). I suggest reviewing the sentence.

Introduction, line 43 e 44 - I agree with the sentence. What properties are we talking about? It is different if we speak of a flexural characterization than a low-velocity impact characterization, for example. Furthermore, for example, to achieve the desired properties we will have to add more layers of one or another type of fiber, it is inevitable. I suggest justifying the sentence with bibliography.

Introduction - I suggest approaching the theme of the characterization made: three-point bending, low-velocity impact and burning based on bibliographical references.

Experimental, line 127 - Why are you choosing these values for the time and temperature variables? On what basis? Previous works? It was important for the reader to understand.

Burning Behavior, line 292 – “Figures 5 and 7”, "7" must be replaced by 9 and the numbering of figures must be revised, as figure 7 does not exist.

Burning Behavior, line 301 - “Table 3”, “3” must be replaced by 2.

Author Response

We highly appreciate all the comments and find them very useful. We agree with the recommendations that the manuscript should be improved, so efforts have been made to correct the article according to the comments. Below we enclose the replies to the comments and recommendations made by the Reviewer. All of the changes in the text are highlighted using colour.

  1. Abstract, line 23 - It is said that the different fabrics of the different fibers used in this study are of "similar grammage".

For example the grammage of the: aramid fabric (A) is 300 g/m2 and flax fabric (F) is 500 g/m2. the difference is obvious for the authors to be able to assert this. This difference is also evident in Fig. 1 (SEM). I suggest reviewing the sentence.

Thank the Reviewer for the valuable comment. We carefully improved that inaccuracy in the manuscript.

  1. Introduction, line 43 e 44 - I agree with the sentence. What properties are we talking about? It is different if we speak of a flexural characterization than a low-velocity impact characterization, for example. Furthermore, for example, to achieve the desired properties we will have to add more layers of one or another type of fiber, it is inevitable. I suggest justifying the sentence with bibliography.

According to the Reviewers comment, additional comments and bibliography have been included in the revised version of manuscript.

  1. Introduction - I suggest approaching the theme of the characterization made: three-point bending, low-velocity impact and burning based on bibliographical references.

The main purpose of the work was to compare various manufacturing techniques. Still, the Reviewer is right, and it was necessary to complete the information about the anticipated changes concerning the measurement techniques used. The revised manuscript version provides an additional subsection in the Introduction part.

  1. Experimental, line 127 - Why are you choosing these values for the time and temperature variables? On what basis? Previous works? It was important for the reader to understand.

We appreciate the Reviewer’s remarks. The manufacturing parameters were selected based on our previous research work. The sentence has been changed accordingly.

  1. Burning Behavior, line 292 – “Figures 5 and 7”, "7" must be replaced by 9 and the numbering of figures must be revised, as figure 7 does not exist.

We thank the Reviewer for the suggestion. The authors revised the numbering of all figures in the manuscript.

  1. Burning Behavior, line 301 - “Table 3”, “3” must be replaced by 2..

The authors wish to thank the Reviewer for drawing attention to the error that was made.

We have also corrected quite a number of other minor errors which we noticed while working on the text, and we believe that our manuscript in the present form can be published in the Journal.

Yours faithfully,

Kamila Salasinska,

Mateusz Barczewski

Reviewer 4 Report

An interesting study was carried out on the effect of fiber hybrid on the mechanical properties and fire behavior of composites through the different preparation technology. In order to improve the quality of the paper, the authors are suggested to consider the following comments for a major revision.

  1. The title does not highlight the hybrid composites. It is suggested to modify the title to be consistent with the content of the full text.
  2. In the part of the abstract, why does the authors study the mechanical properties and fire behavior of hybrid composite? It is recommended that the research background and application field should be given at the beginning. In addition, the authors have adopted several manufacturing methods to prepare hybrid composites. Why does it not include pultrusion production process? As knows, the pultrusion technology is very mature, which can prepare composite plates with stable and excellent properties.
  3. Line 47-82, the detailed production process of composite materials is summarized. However, it is puzzling that the pultrusion production technology is not included in this part. It is suggested to add this part and compare with other production methods to illustrate the advantages and disadvantages.
  4. In the present paper, different fibers are adopted to prepare the hybrid composites. However, the effects of fiber hybrid mode, hybrid mechanism on short-term performance (mechanical properties) and long-term performance (durability) are more attractive to readers. Unfortunately, the authors did not make a full and comprehensive summary of this part. It is suggested that the authors review the following research work on hybrid composites and make a necessary supplement. https://doi.org/10.1080/15376494.2021.1974620. Journal of Materials Research and Technology, 2021, 14:2812-2831. Composite Structures, 2021, 261:113285. Construction and Building Materials, 2022, 314:125587.
  5. In Table 1, the authors adopted five kinds of fibers to prepare the hybrid composites. What is the significance of choosing five fibers for hybrid? In addition, what are the selection principles for the laminate stacking sequences?
  6. It is suggested that the authors provide the pictures of hybrid plates prepared by 6 different schemes and manufacturing method. Furthermore, the effects of different preparation methods on the surface morphology of the plate can be compared.
  7. Part 2.2, it is recommended that the performance test and microanalysis (SEM) were written logically. Macro performance tests should be presented at first.
  8. Figure 1 shows the SEM images of brittle-fractures of hybrid composites. This part should be put after the mechanical property evaluation. Because SEM analysis was based on the fracture after the mechanical properties. In addition, the size of Figure 1 is too large and is recommended to adjust it.
  9. In figures 2 and 4, flexural strength and impact properties are different for different hybrid composites (for example, type 2 has the highest flexural strength and type 3 has the higher energy absorption). How to balance the above two properties to obtain the best material?

Author Response

We highly appreciate all the comments and find them very useful. We agree with the recommendations that the manuscript should be improved, so efforts have been made to correct the article according to the comments. Below we enclose the replies to the comments and recommendations made by the Reviewer. All of the changes in the text are highlighted using colour.

  1. The title does not highlight the hybrid composites. It is suggested to modify the title to be consistent with the content of the full text.

We thank the Reviewer for the remark. The title has been changed accordingly.

  1. In the part of the abstract, why does the authors study the mechanical properties and fire behavior of hybrid composite? It is recommended that the research background and application field should be given at the beginning. In addition, the authors have adopted several manufacturing methods to prepare hybrid composites. Why does it not include pultrusion production process? As knows, the pultrusion technology is very mature, which can prepare composite plates with stable and excellent properties.

We thank the Reviewer for comment. The abstract has been revised according to comments from all reviewers. Moreover, the revised manuscript version provides an additional subsection in the Introduction part.

The work carried out is part of a project related to producing thin-walled composites for the railway industry with improved fire retardancy. The target element is products in the form of boards; therefore, the shaping technologies were considered concerning the performance conditions of the industrial partners. While a pultrusion technology is commonly used to produce continuous elements in the form of bars or profiles, the formation of highly filled with flame retardants and particle-shaped fillers plates would be challenging to proceed correctly. The comment is valuable from the point of view of analyzing the mechanical properties of hybrid composites; however, the presented case study did not consider the application of novel technology. Unfortunately,  there is no possibility to acquire the equipment needed to carry out the pultrusion process of thermoset polymers. Therefore, even if the concept of developing research into products by this method would be justified from a scientific point of view, it cannot be implemented at the current stage. Nevertheless, we would like to thank the Reviewer for this remark because perhaps it will provide a foothold for further work in cooperation with another unit, which will allow expanding our knowledge in the field of hybrid composites developed by our team.

  1. Line 47-82, the detailed production process of composite materials is summarized. However, it is puzzling that the pultrusion production technology is not included in this part. It is suggested to add this part and compare with other production methods to illustrate the advantages and disadvantages.

According to the Reviewers' comment, an additional description was added in the reviewed version of the manuscript. It justifies the selection of the considered techniques and corresponding to why the pultrusion technique was not used. While Pultrusion is an exciting technology to obtain products with exceptional mechanical performance, they are characterized by a greater or high degree of unidirectional distribution of the fibers. In our case, the technology selection was related to the necessity to later produce composites for the railway industry, which complex shapes and a large surface will characterize. It is also essential to obtain a limited anisotropy of mechanical properties in the laminate plane. Therefore, the technique of pultrusion has not been considered in this paper.

  1. In the present paper, different fibers are adopted to prepare the hybrid composites. However, the effects of fiber hybrid mode, hybrid mechanism on short-term performance (mechanical properties) and long-term performance (durability) are more attractive to readers. Unfortunately, the authors did not make a full and comprehensive summary of this part. It is suggested that the authors review the following research work on hybrid composites and make a necessary supplement. https://doi.org/10.1080/15376494.2021.1974620. Journal of Materials Research and Technology, 2021, 14:2812-2831. Composite Structures, 2021, 261:113285. Construction and Building Materials, 2022, 314:125587.

The authors would like to thank the Reviewer for this valid comment. In the case of the description of both mechanical tests, the additional comments and a summary with the use of relevant literature references were introduced in the revised manuscript version. Additional references have been included in the revised manuscript version.

  1. In Table 1, the authors adopted five kinds of fibers to prepare the hybrid composites. What is the significance of choosing five fibers for hybrid? In addition, what are the selection principles for the laminate stacking sequences?

The results presented in the article were carried out as part of the first stage of the 3-years project, the aim of which was to select the manufacturing method. Based on the literature data and team members' experience, several different reinforcements in the form of fabrics and powder fillers were selected. The task of the next stage will be the selection of the composition and its optimization determined by the application. The stacking sequence was based on our preliminary research. The article [Salasinska, K.; Barczewski, M.; Aniśko, J.; Hejna, A.; Celiński, M. Comparative Study of the Reinforcement Type Effect on the Thermomechanical Properties and Burning of Epoxy-Based Composites. J. Compos. Sci. 2021, 5, 89] presents the results of mechanical and burning tests for composites with the same fabrics but one type of reinforcement. The results of the tests carried out using the cone calorimeter were decisive in selecting the sequence of fabrics. The compositions presented in this article are only a fragment of the research to show established relationships and constitute an introduction to the following works.

Thanking the Reviewer for the remark, we would like to say that the explanation has been introduced to the article.

  1. It is suggested that the authors provide the pictures of hybrid plates prepared by 6 different schemes and manufacturing method. Furthermore, the effects of different preparation methods on the surface morphology of the plate can be compared.

A macroscopic assessment of the sample's surface was made, and the photos are presented in the supplementary data. In the text, referring to the limited saturation of the top layer, a reference to the appointed photos was made. The authors would like to the Reviewer for the valuable comment, and this is an excellent supplement to the observations made by the authors.

  1. Part 2.2, it is recommended that the performance test and microanalysis (SEM) were written logically. Macro performance tests should be presented at first.

We would like to admit a mistake in the article. The presented SEM images refer to cross-sections, not breakthroughs, as stated in the figure captions. The article has been updated. The authors apologize for any inconvenience caused and state that the scientific conclusions are unaffected.

We appreciate the Reviewer’s remark; however, since the SEM images show the primary structure of the composites after their production, but before mechanical testing, the authors prefer the original order.

  1. Figure 1 shows the SEM images of brittle-fractures of hybrid composites. This part should be put after the mechanical property evaluation. Because SEM analysis was based on the fracture after the mechanical properties. In addition, the size of Figure 1 is too large and is recommended to adjust it.

We would like to admit a mistake in the article. The presented SEM images refer to cross-sections, not breakthroughs, as stated in the figure captions. One of the co-authors likely made this error at the last reading stage and was not noticed before submitting the article to the journal. The article has been updated. The authors apologize for any inconvenience caused and state that the scientific conclusions are unaffected.

Thanking the Reviewer for the remarks, we would like to say that the changes concerning the size of the Figure have been made.

  1. In figures 2 and 4, flexural strength and impact properties are different for different hybrid composites (for example, type 2 has the highest flexural strength and type 3 has the higher energy absorption). How to balance the above two properties to obtain the best material?

An additional summary of the evaluation of mechanical properties is presented in the revised version of the manuscript according to reviewers' comments.  

We have also corrected quite a number of other minor errors which we noticed while working on the text, and we believe that our manuscript in the present form can be published in the Journal.

Yours faithfully,

Kamila Salasinska,

Mateusz Barczewski

Round 2

Reviewer 4 Report

After a full preview, I found that the authors have made comprehensive and sufficient modifications according to the comments of the reviewers. Therefore, I recommend accepting the paper in the present form.